# Characterization and Analysis of the Mortars of the Church of Santo Domingo in Quito (Ecuador)

M. Lenin Lara Calderón [1,2,*] , David Sanz-Arauz [1], Sol López-Andrés [3] and Inés del Pino [4]

1. Department of Construction and Architectural Technology, Polytechnic University of Madrid, 28040 Madrid, Spain
2. Faculty of Arts, Design and Architecture, UIDE International University of Ecuador, Simón Bolívar Av., Jorge Fernández Av., Quito 170411, Ecuador
3. Department of Mineralogy and Petrology, Complutense University of Madrid, 28040 Madrid, Spain
4. Faculty of Arts, Design and Architecture, Pontifical Catholic University of Ecuador, Quito 170121, Ecuador
* Correspondence: lenin.lara.calderon@alumnos.upm.es; Tel.: +593-994384851

**Abstract:** The religious art of the Dominican order is reflected in Santo Domingo Church, which was built between 1541 and 1688. This work of heritage architecture, one of the first to be built in the colonized city, was affected by multiple earthquakes, interventions, and constructions that have not been clearly recorded. A total of 13 samples were taken from the mortar inside the cloister, central nave, and side chapel, following the minor destruction-testing protocols and standards suggested by the research team. The analysis included mineral characterization studies and quantitative analysis by X-ray diffraction, petrographic, and scanning electron microscopy with microanalysis of the samples. The results showed the presence of volcanic aggregates and lime mortars, mortars of rustic composition and coarse manufacture. The results of mineralogical data and texture have allowed us to corroborate the historical information described by the chroniclers, to date relatively studied sites and to establish a hypothesis of constructive stages.

**Keywords:** lime mortar; mortars with volcanic aggregate; mineralogy of historic mortars; Quito Cultural Heritage; Church of Santo Domingo of Quito

## 1. Introduction

By December 1534, the original city of Quito was occupied by Spaniards, and during the colonial period, it was the political and economic center of the new fief. The delimitation of the plots was made based on the central axis, which was the main square; this is where the buildings of the royal and religious authorities were placed. At the limit of this node, a plot of land was given to the Dominican Order to spread the Christian faith and religion among polytheistic and pagan natives. This ancient center houses pre-Hispanic and Hispanic material and immaterial wealth in its 375.25 hectares with more than 4674 built properties, of which 130 are monumental properties. In the sixteenth century, there were 15 religious buildings with a central consolidated core estimated at 70.43 hectares [1]. Quito was recognized by UNESCO in 1978 [2,3] for having the best preserved and least altered historical center of Latin America (Figure 1). Here, the church, the convent and the square of Santo Domingo can be found [4].

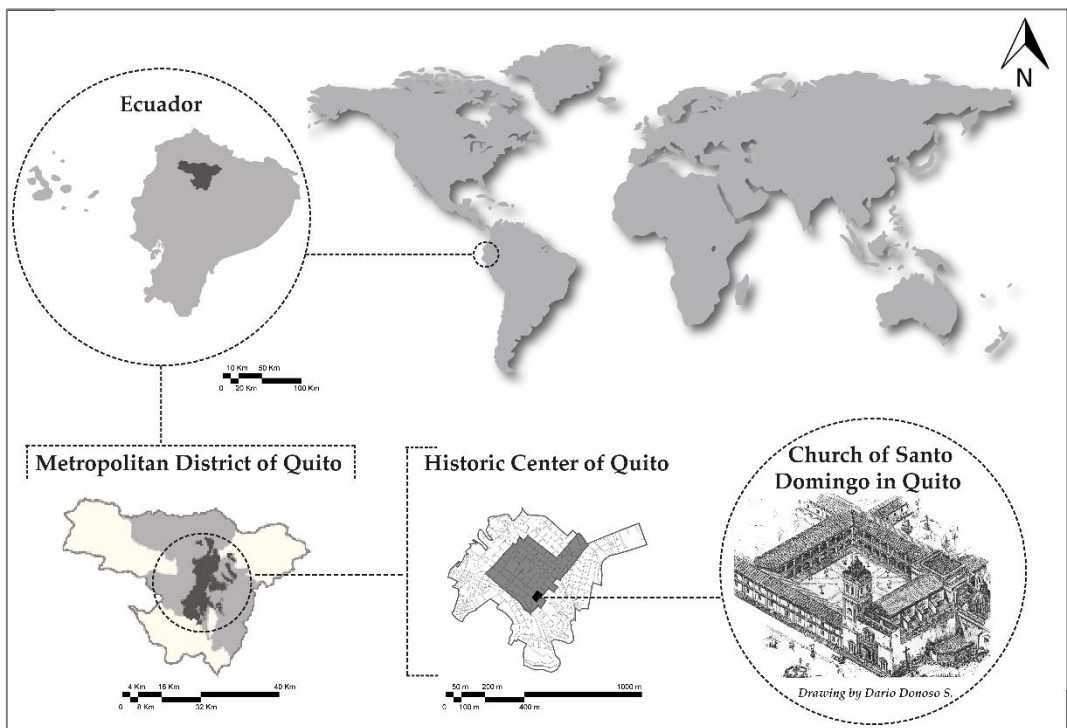

**Figure 1.** Location and context of the studied area. Drawing by Dario Donoso Samaniego, on page 139 [5].

The Dominicans were established in the Viceroyalty of Peru and Quito, by then considered a circumscription of Cusco. However, in the process of expanding their faith, they were the first missionaries to travel the coasts of this territory to expand their Christian doctrine [6]. The royal decree of 1563 named Quito a city [4,7], and the Dominicans obtained this land to settle on the border of the central nucleus and the Loma Grande.

*Constructive Historical Chronology of the Temple*

The topographical conditions and accessibility to the property established that, between 1541 and 1580, the Dominican order built a provisional convent made of cedar wood and carved coffered ceilings (shown to Figure 2). It was located between the current courtyard of the main cloister and San Fernando College, according to studies by Josef Buys [8]. At the beginning of 1581, Francisco Becerra, who previously worked building temples in Mexico and Lima, began drawing the blueprints of the church and convent with serious modifications of the previous cloister with more generous dimensions for the faithful and religious, orienting it towards the south of the site [4,9].

The temple was developed with a single central nave, apse, and vaulted side chapels. On the right side of the transept, the Chapel of the Rosary was built, which houses an original altarpiece from the sixteenth century, with Baroque detail and samples from several artists. Among the artists that stand out are Priest Pedro Bedón and Artist Miguel de Santiago, as well as member artists of the native brotherhood of the Dominican church of the Rosary, Nicolás Javier Goríbar, Andrés Sánchez Galque, Alonso Chacha, Francisco Gocial, Jerónimo Vilcacho, Juan Diez Sánchez, Sebastián Gualoto, Francisco Guajal, and Juan Greco Vásquez, among others [10].

After Becerra's work, the remaining activities were handed to Fray Rodrigo de Lara Manrique, who followed the guidelines indicated previously. Later, Fray Antonio Rodríguez took charge of the convent's work, while Fray Juan Mantilla was in charge of concluding the church's works on 15 January 1688 [7,11]. By 15 July 1688, an official report was issued by Antonio Rodríguez on the conditions of the convent. San Fernando

College was inaugurated on 6 August 1688, for poor children from Loma Grande, totally free of charge.

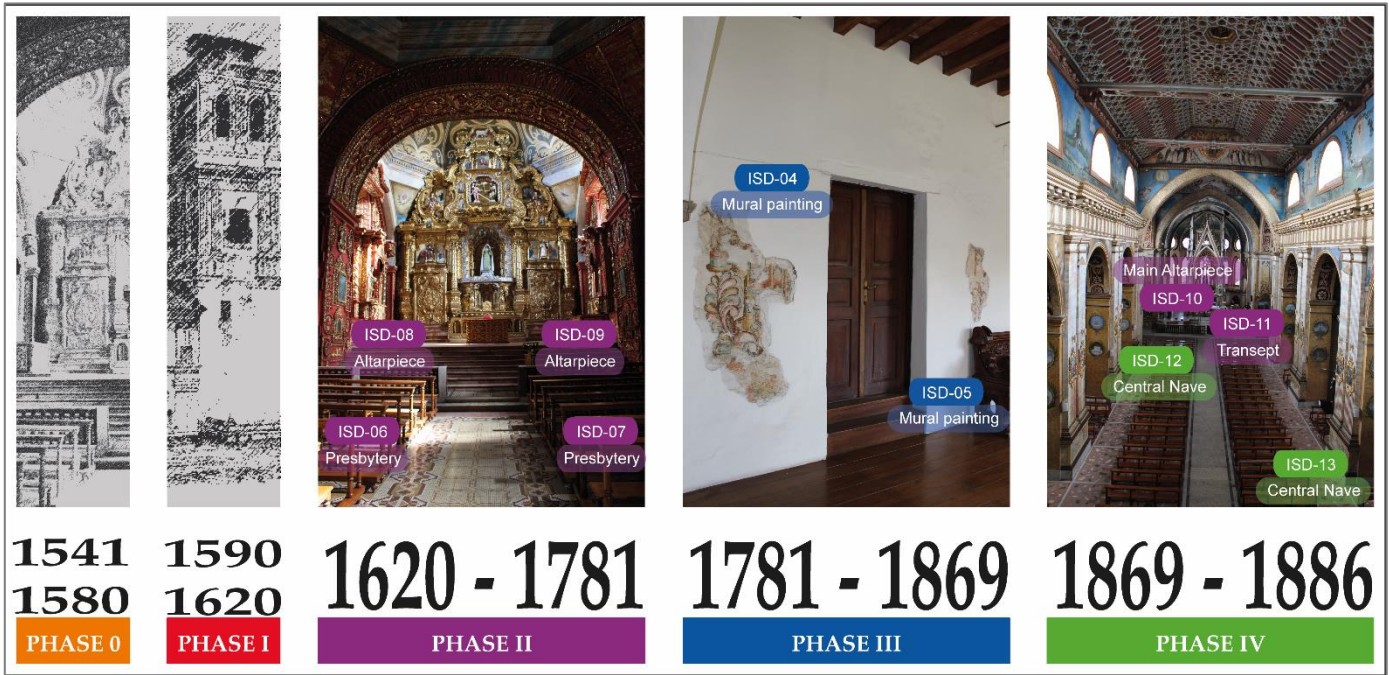

**Figure 2.** Image of the five chronological phases of the construction of the Santo Domingo church in 345 years. (1541–1580) Earth construction; (1590–1620) Phase I; (1620–1781) Photography of the Chapel of the Rosary; (1781–1869) Photography of the mural in the convent corridor; and (1869–1886) Photography from the central nave to the main altar, with the location of the places where the mortar samples were extracted.

The Dominican complex has suffered several earthquakes that have affected its structure, among which we highlight the one in 1587 where part of the church collapsed and the tower was damaged. In 1736, the tower and roofs were damaged. In 1751, the bell tower and the bell itself were broken. In 1755, due to what is known as the "Quito earthquake", the church was severely affected, causing the collapse of the walls and ceilings. In 1787, there was a collapse and sinking of the church and the novitiate. In 1797 there was a partial collapse of the tower, and there were damages to the novitiates. Finally, in 1859, the tower was damaged and the support was destroyed, which is why the walls were raised to prevent further collapse [12,13].

Between 1869 and 1886, a group of Italian Dominicans made important reforms inside the temple, and a new spatial conception of the convent and its novitiate was projected. The plain façade of the church did not suffer changes by reformers, but there are contrasts between the Renaissance style of its exterior and the Baroque style of its interior, where an ornamental and monochromatic style prevail. In the twentieth century, between 1990 and 1996, restoration work was carried out on the altarpiece and the choir loft, and from 1998 to 2002, work was carried out on the frame (roof trusses) and ceilings [8].

Information regarding previous research related to heritage mortars is limited. The bilateral project between Ecua-Bel and the National Institute of Cultural Heritage (INPC from its initials in Spanish), coordinated by Patrick de Sutter and Marcela Alemán, had as an objective the study of the mural painting in 1993 [14]. Later, the Rescue Fund of the Municipality of the Metropolitan District of Quito (FONSAL from its initials in Spanish), through a private consultancy by Paulina Moreno in 2006, again conducted studies on mural painting of the transept of Santo Domingo [15].

Regarding work related to building materials, we can highlight Moropoulou's work, who conducted research on the San Francisco monastery in adobe and brick masonry [16], as well as research on the characterization of mortars of the Church of the Company of Jesus [17]. This investigation focused on the types of mortar used and the dosage, compositions, and textures. of heritage samples [18]. Due to the value and historical importance of the coatings used in this temple, we had to determine, in addition to those mentioned above, their chemical, physical, and mechanical structure [19,20]. The use of different microscopic techniques as a fundamental aspect of data processing allowed us to visualize the precise characterization of binders, aggregates, additives, inorganic mixtures, etc. [21–23], which were used in these historical mortars, and therefore the team was able to validate the sequential historical chronology described.

Historical documents show that the Real Audiencia of Quito decided that Dominicans obtained the material needed to build their properties from the lime quarries of the Panzaleo sector, on the southern border of the province [24]. This Andean region has the main mineral components of the volcanic rock used as aggregate in these mortars [25,26].

This work is part of a larger research project in which chemical–mineralogical properties will be determined. In this paper, our aim is to identify, in a technical manner, the mineralogical composition of the mortars used in the construction of the Santo Domingo of Quito church and thus improve the level of future interventions in this building or in those that maintain the same material.

## 2. Materials and Methods

### 2.1. Materials

For this work, 13 mortar samples (shown to Figure 3) were collected from the Santo Domingo church, considering the indications made by temple technicians about the places that presumably have not been altered in the rehabilitation stages. To carry out the sampling, three sectors were identified depending on the construction phase.

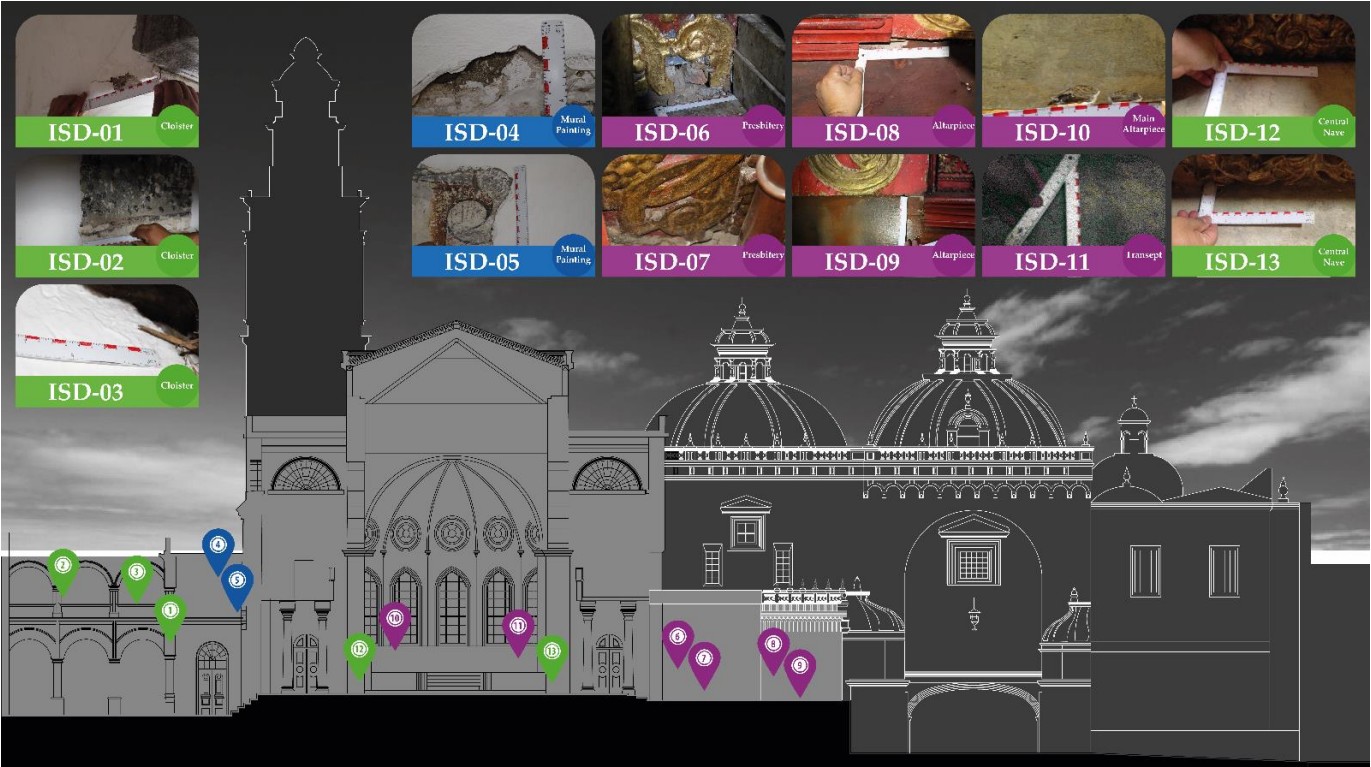

**Figure 3.** Blueprints of the church of Santo Domingo of Quito with the location of the analyzed samples; the figure describes the 13 samples with their locations.

The church of Santo Domingo historically records a temporary construction of adobe (1541–1581), and later Phase I (1590–1620), of which we do not have samples.

For Phase II (1620–1781), the samples were coded as ISD-06, ISD-07, ISD-08, ISD-09, ISD-10, and ISD-11. Samples 6, 7, 8, and 9 were extracted from the Chapel of the Virgin of the Rosary, sample 11 was extracted from the south surface of the transept, and sample 10 was part of the nineteenth century reconstruction of the novitiate.

In Phase III (1781–1869), samples coded as ISD-04 and ISD-05 were extracted from the surface of the wall of the convent cloister adjacent to the north wall of the temple.

For Phase IV (1869–1886), samples coded as ISD-01, ISD-02, ISD-03, ISD-12 and ISD-13 are identical to those of the Italian Reformation that intervened in the cloister of the convent (samples 1, 2, and 3) and the Central Nave of the Temple (samples 12 and 13).

It is important to consider that previous historical studies record dated zones. For example, there are records from 1631 about the surface of the mural painting of the transept 'near *our sample 11*' and a record from 1789 about the surface of the mural painting of the cloister 'near *our samples 4 and 5*' [10,15].

### 2.2. Methods

Combined methodology that collects qualitative, quantitative, and analytical data was used in a systematic way that began with a historical and constructive analysis of the place where the samples were taken, (shown to Figure 4).

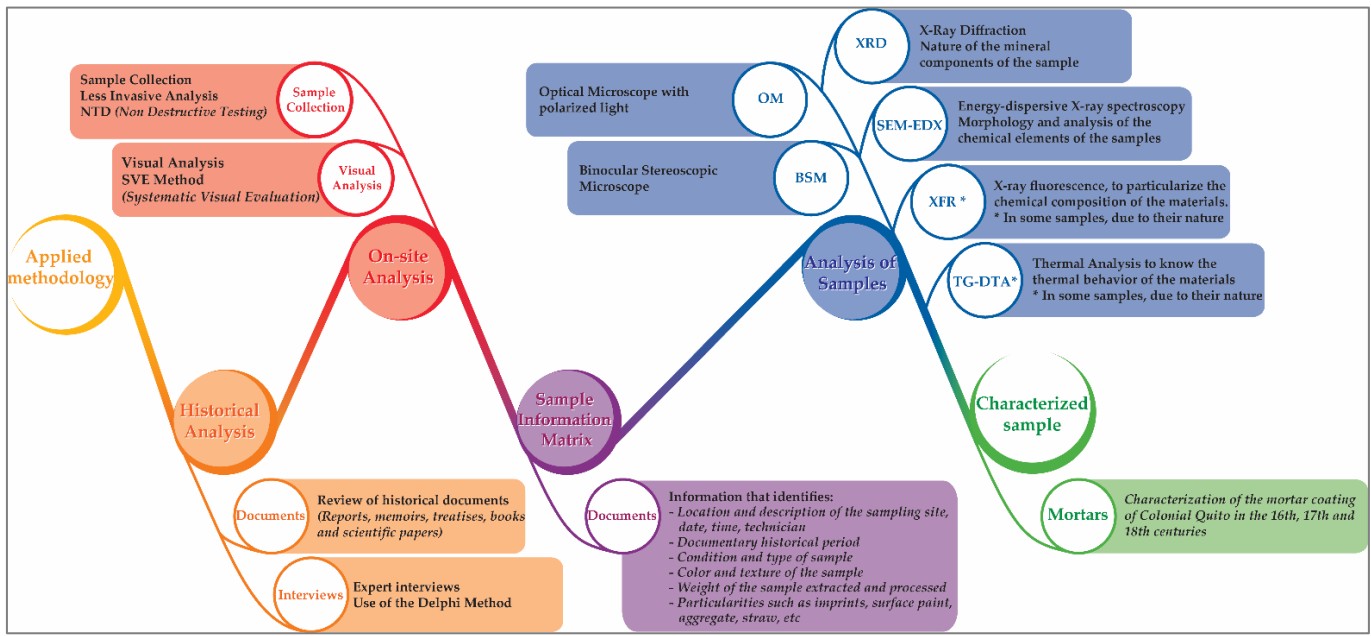

**Figure 4.** Description of the methodological process generated for the analysis of samples from general research. * Analyses performed for some of the samples due to their nature.

Historical research requires knowledge about the historical process from the time of construction to the moment of study. This research analyzes the historical context, knowing its age, the construction system, the historical destruction during the construction phase, and the main transformations the building underwent due to earthquakes or anthropic effects [27,28].

We studied existing historical and graphic documents such as photographic images, planimetric drawings, and other treatises in which historical information is available about the construction systems, materials, and techniques used, with an emphasis on interviews with researchers, historians, technicians, and knowledgeable experts about the subject using Delphi methodology.

Initially, once on site, the SVE method (Systematic Visual Evaluation) was used. This requires having a dynamic protocol that begins with knowing the site, relating the historical part that was previously classified with the perceived data of the place where the external conditions that affect the building are studied, such as its orientation, sunlight, prevailing winds, precipitation, and rain, all of which affect the construction materials that are part of the study and affect the research.

In the interior part of the building and with the existing description of the historical element by construction stages and the results of the method described above, we visually identify the type of coating that can be used as part of the study and locate the possible points of sample extraction, for which we use the least aggressive tests on the heritage asset, using the methodology for the sampling of materials of cultural heritage classified by UNE-EN 16085 [18].

It should be considered that the sample should not be visually contaminated with differences due to the effects of predecessor construction pathologies of a physical, mechanical, chemical, or biological nature such as humidity, detachment, efflorescence, or erosion, among others.

At this point, an information matrix was generated to identify and locate the place from where the sample was taken, the historical documentary period to which it belongs, and the weight of the extracted sample, as well as the quantity of material to be processed and its color and texture, and to identify the particularities or imprints left by external elements such as surface painting, aggregates, straw, etc. Subsequently, it was determined which and how many significant samples must undergo archeometric analysis to identify and characterize them [19,25,26].

For characterization, we started with a binocular stereoscopic microscope that allows us to gain a stereoscopic view of the sample, and a Zeiss Stemi 305 instrument was subsequently used, both for identification by X-ray diffraction (XRD) and for mineralogical characterization of the 13 historical samples. A 10 to 20 gr portion was extracted and ground in an agate mortar and sieved using the polycrystalline powder method, which requires sample fractions with grains smaller than 53 μm [29–32]. Mineralogical identification was carried out by XRD [33,34] on a Bruker D8 ADVANCE diffractometer with Cu radiation, where disoriented powder diffraction patterns were obtained in an angular interval between 2 and 65°, in 1 s, with a step size of 0.02°. The relative proportions of each mineralogical phase were identified and determined following the Chung method (1975) [35], and using Bruker's EVA software, maintaining the experimental errors of the method plus or minus 5%.

Thin sections of the samples ISD-08 and ISD-11 of phase II, ISD-05 of phase III, and ISD-12 of phase IV were prepared for Petrographic Microscopy (PM) using Zeiss Primotech model equipment and for morphological and chemical study using a scanning electron microscope (SEM-EDX) [23,36,37]. The equipment used was JEOL JSM-820, with a secondary and backscattered electron detector and microanalysis. The software used for the acquisition, treatment, and evaluation of the analyses was EDX Oxford ISIS-Link. The analysis was carried out by the Geological Techniques Unit of the Support Center for Research in Earth Sciences and Archaeometry of the Complutense University of Madrid (Spain).

## 3. Results

Table 1 allows us to identify the mineralogical composition and semi-quantification (+5%) resulting from the use of the XRD technique applied to identify the crystallographic structure in materials science. However, for this purpose, the samples have been grouped chronologically according to the historical information used in the investigation.

**Table 1.** XRD mineralogical identification and quantification. The symbols used for the mineralogical phases are given according to [38]. Quartz = Qz, plagioclase = Pl, amphibol group = Amp, calcite = Cal, Illite group (ClayMinerals) = Ilt, and larnite (belite) = Lrn.

| Samples | | | | | | |
|---|---|---|---|---|---|---|
| | **Qz** | **Pl** | **Amp** | **Cal** | **Ilt** | **Lrn** |
| **ISD-06** | 3 | 87 | 6 | | 4 | |
| **ISD-07** | <1 | 86 | 12 | | 2 | |
| **ISD-08** | | 57 | 22 | 18 | 3 | |
| **ISD-09** | 1 | 70 | 18 | 10 | 1 | |
| **ISD-10** | <1 | 88 | 6 | 5 | 1 | |
| **ISD-11** | 3 | 84 | 9 | | 4 | |
| **ISD-04** | 2 | 64 | 12 | 6 | 4 | 11 |
| **ISD-05** | 2 | 80 | 7 | 8 | 3 | |
| **ISD-01** | 2 | 74 | 18 | 4 | 2 | |
| **ISD-02** | | 92 | 2 | 3 | 3 | |
| **ISD-03** | 3 | 54 | 7 | 30 | 6 | |
| **ISD-12** | 2 | 80 | 13 | | 5 | |
| **ISD-13** | 2 | 83 | 9 | | 6 | |

The mineralogical identification carried out by XRD and petrographic microscopy (the polished thin section) allows us to present the whole set of analytical qualitative data and the manufacture of mortars. For this study, it should be considered that the sample ISD-04 is contaminated with synthetic elements that do not belong to the temporality of the study, so it was discarded. Table 2 describes the types of mortars found in the three construction phases that are part of the chronology examined.

**Table 2.** Type of mortars found in the church of Santo Domingo.

| Samples | Mortar |
|---|---|
| **ISD-06** | Volcanic aggregate |
| **ISD-07** | Volcanic aggregate |
| **ISD-08** | Lime mortar with volcanic aggregates |
| **ISD-09** | Lime mortar with volcanic aggregates |
| **ISD-10** | Lime mortar with volcanic aggregates |
| **ISD-11** | Volcanic aggregate |
| **ISD-04** | Intervened area *"cement mortar"* |
| **ISD-05** | Lime mortar with volcanic aggregates |
| **ISD-01** | Lime mortar with volcanic aggregates |
| **ISD-02** | Lime mortar with volcanic aggregates |
| **ISD-03** | Lime mortar with volcanic aggregates |
| **ISD-12** | Volcanic aggregate |
| **ISD-13** | Volcanic aggregate |

To identify the morphology, we used polarized optical microscopy (POM) and SEM to understand the different mineral phase textural relationships. XRD was used for the mineralogical identification. The samples ISD-08 and ISD-11 from phase II (Figures 5 and 6), ISD-05 from phase III (Figure 7), and ISD-12 from phase IV (Figure 8) were chosen as representative samples of the use of volcanic aggregates in different ornamental situations. The use of these in stucco-type finishes is worth mentioning [39,40].

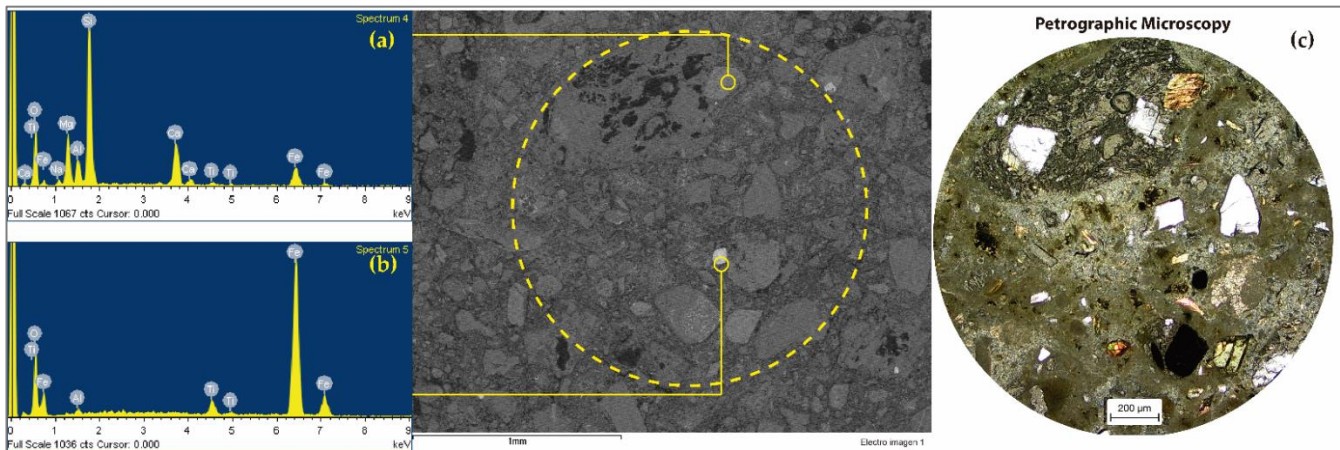

**Figure 5.** Sample ISD-08 (**a**,**b**) chemical analysis by EDX-SEM supported by a BSE image of lime mortar with volcanic aggregates, and (**c**) thin section image at $50\times$ magnification under PM with X polarizer.

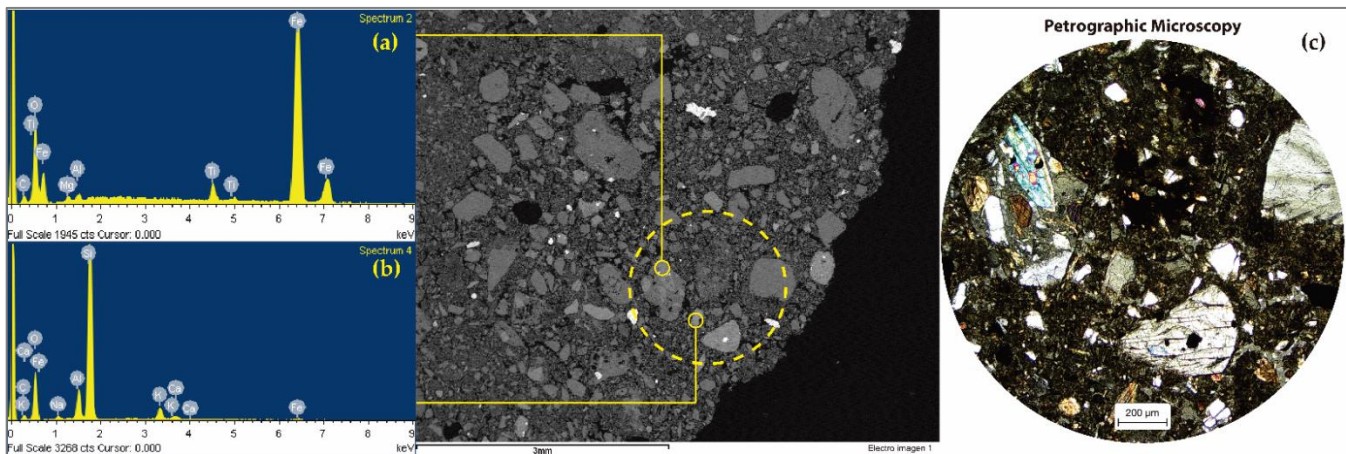

**Figure 6.** Sample ISD-11 (**a**,**b**) chemical analysis by EDX-SEM supported by a BSE image of lime mortar with volcanic aggregates, and (**c**) thin section image at $20\times$ magnification under PM with X polarizer.

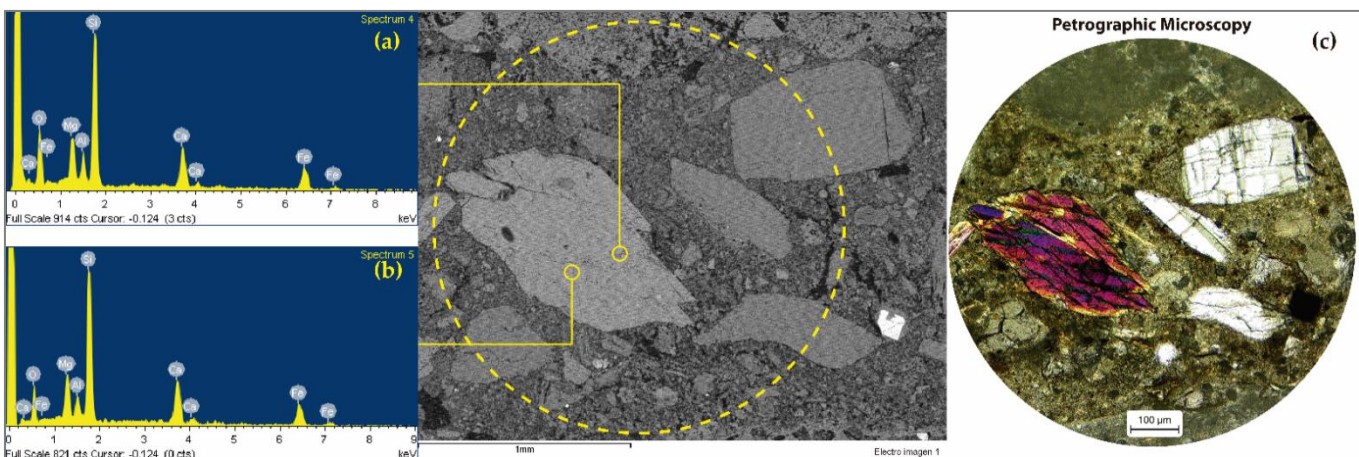

**Figure 7.** Sample ISD-05 (**a**,**b**) chemical analysis by EDX-SEM supported by a BSE image of lime mortar with volcanic aggregates, and (**c**) thin section image at $150\times$ magnification under PM with X polarizer.

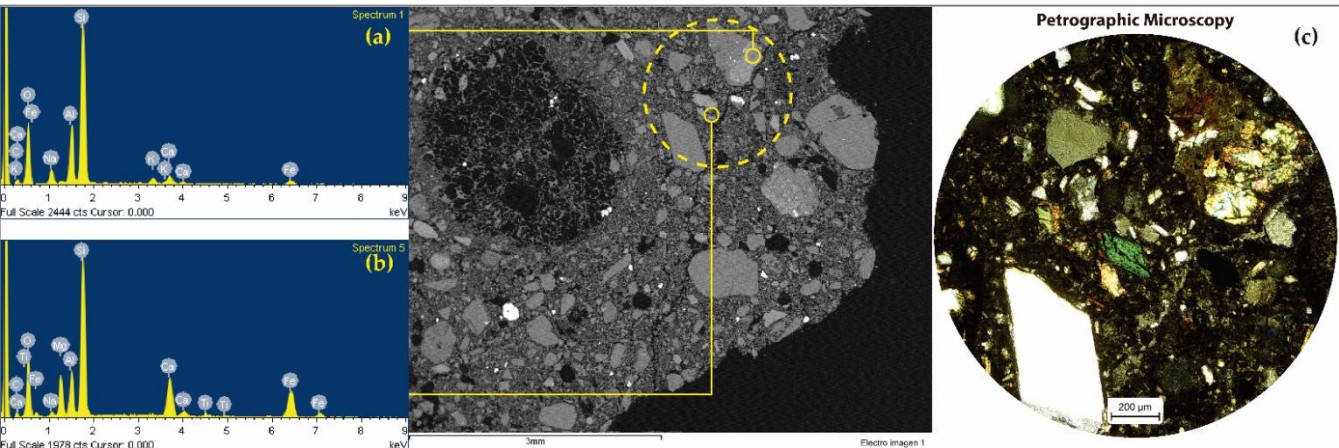

**Figure 8.** Sample ISD-12 (**a**,**b**) chemical analysis by EDX-SEM supported by a BSE image of lime mortar with volcanic aggregates; and (**c**) thin section image at 50× magnification under PM with X polarizer.

As shown in (Figure 5) of sample ISD-08, this EDX analysis was performed at nine different points of the main chemical elements. We have identified relevant points: (a) spectrum 4 with Si, Mg, Ti, Ca, Al, and Fe; (b) spectrum 5 with Fe, Ti, and Al; (c) an image obtained at 200× magnification of the petrographic microscope that allows visualizing the properties of double refraction or crystallographic directions of the sample of the Chapel of the Rosary by controlling the contrast of the image and color variation.

In the case of sample ISD-11 (Figure 6), this EDX analysis was performed at six different points of the main chemical elements, and we have pointed out some relevant points: (a) spectrum 2 with Fe, Ti, Al, and Mg; (b) spectrum 4 with Si, Al, Ca, K, and Na; (c) an image obtained at 200× magnification of the petrographic microscope that allows visualizing the properties of double refraction or crystallographic directions of the sample from the lower part of the transept by controlling the contrast of the image and color variation.

EDX analysis of sample ISD-05 (Figure 7) was performed at nine different points of the main chemical elements. We have pointed out some relevant points: (a) spectrum 4 with Si, Mg, Ca, Al, and Fe; (b) spectrum 5 with Si, Mg, Ca, Al, and Fe; (c) an image obtained at 500× magnification of the petrographic microscope that allows visualizing the properties of double refraction or crystallographic directions of the sample of coating of mural painting by controlling the contrast of the image and color variation.

Finally, EDX analysis of sample ISD-12 (Figure 8) was performed at nine different points of the main chemical elements, and we have pointed out relevant points: (a) spectrum 1 with Si, Fe, Al, Ca, Na, and K; (b) spectrum 5 with Si, Mg, Ti, Al, Fe, Na, Ca, and Ti; (c) an image obtained at 200× magnification of the petrographic microscope that allows visualizing the properties of double refraction or crystallographic directions of the sample of the central nave of the temple by controlling the contrast of the image and color variation.

## 4. Discussion

The identification, characterization, and structure of several of the elements studied in the Santo Domingo church allow us to determine the mineral composition of the volcanic origin as lime mortar with volcanic aggregates. Here, a rustic and rudimentary manufacture of the aggregate is identified, allowing evidence of its vitreous texture. This is evident in many of the samples that are part of the study, which endured superficial whitewashing of the interior of the spaces as a feature of prestige in the architecture of the time, allowing the surface to ornament that area and the asepsis of the enclosure.

Our analyses identified quartz with minimal proportions in all mortars, between 1% and 3%, and calcite in percentages between 3% and 30%, except in the samples ISD-06,

ISD-07, ISD-11, ISD-12, and ISD-13, in which calcite was not identified at all. It should also be noted that the presence of plagioclase and hornblende-type amphibole with significant values was observed in all samples. These are the main components of the volcanic aggregate found. The Andean valley has a dozen ancient andesitic volcanic centers, characterized by basaltic andesite (57–60% $SiO_2$ and 0.5–1.0% $K_2O$) where, according to historical data, the extraction quarry of the Dominican religious order was located [23,41,42].

The aphanitic or fine-grained texture is evident in the microscopic analysis of the samples where the vitreous crystals expose fragments of andesite and dacite manufacture. These materials are of widespread use in the civil and religious constructions of the Historic Center of Quito. This is evident in the studies of the Church of the Company of Jesus and the Church of San Francisco [16,17].

In addition to the presence of anorthite in the triclinic section, a variety of plagioclases from the feldspar group and muscovite from the silicate group, the phyllosilicate subgroup, were used as secondary minerals. According to the SEM observations, the aggregates did not show any pozzolanic reaction rings in the interface zone.

In the sample ISD-04, larnite (belite) $Ca_2SiO_4$ was identified, granular with a transparent semivitreous structure and anhedral–subhedral crystals that unfold in the matrix of the silicate family, which is a "natural analogue of the synthetic β modification of $Ca_2SiO_4$, which is an important component of portland cement" [43]. It should also be noted that the temple underwent some changes after the Italian Reformation, which ended in 1886 and in the twentieth and twenty-first centuries with several administrations of the enclosure. This generated necessary maintenance interventions, but not well-controlled procedures, resulting in the use of replacement materials with a lack of the original patterns of restoration, causing a gradual deterioration of our heritage.

The samples have different mineral compositions that allow us to identify different typologies, typical of the construction and chronological periods of the temple, generating a mortar based on the volcanic aggregate identified in phase II and phase IV, while sample ISD-05 is a lime mortar with volcanic aggregates, emphasizing that a rough manufacturing scheme of the volcanic compound is evidenced, which would allow us to discuss some aspects of the masonry technique in the construction of the temple.

The analysis obtained by the SEM interaction of the electron beam on the surface of the thin section allowed us to evidence volcanic slag, and thus the validation of the origin of the materials used in the construction of the temple. Therefore, it confirms the hypothesis of the origin of the stone material extraction zone given to Dominicans in the south of the Pichincha province in the Panzaleo sector [24].

When viewing the characteristics of the aggregate, the question of the application of a widespread masonry technique in our region to buildings of historical value arises, which is the crushing of the disassembled mortar and its reintegration with the support of a conglomerating agent, which in this case is lime. With time, the mortar that at first was a rustic monolayer with the addition of lime, either in the mixture or on the surface of the finished coating, acquires greater durability, hardness, workability, and impermeability and improves the appearance of the final finish [25].

## 5. Conclusions

After the structure analysis in the binocular stereoscopic microscope, the samples can be identified by the crystalline contents of the mineral on the petrographic microscope to characterize it by both XRD and SEM-EDX. All of these analyses determined the volcanic origin of the materials used in the construction of temple coatings, confirming that the Panzaleo sector, in the equatorial Andean zone, destined for the Dominican order, was the site of extraction of stone aggregate for the manufacture of mortars [44].

The data obtained by XRD and SEM-EDX of the samples determined common patterns in the components, including calcite, plagioclase as anorthite, clay minerals, and quartz. All of these allowed us to identify some characteristics of the construction period of the church associated with historical facts related by chroniclers that in several cases altered

the original composition of the temple; previous studies had dated some areas that were part of the research.

The samples have a relative presence of conglomerate, lime identified as calcite, except for the ISD-04 sample, which contains larnite (belite). The presence of larnite indicates the use of a cement-based binder used in a subsequent intervention.

Finally, the mortar composition of the Santo Domingo church has its roots in the manufacture of volcanic aggregates and was part of all the construction phases investigated. It should be noted that despite the reforms of the temple, typical of the passing of time, and the seismic past that the church has survived, including several earthquakes (highlighting those of 1755, 1797, 1859 and 1868), its state of conservation is relatively limited. In the visual analysis of the investigation, there were zones from which samples were not extracted due to capillary humidity, bulging, and erosion. Other areas were modified with paint, resins and varnishes that did not appear to be very well implemented, and altered areas with cement mixtures were not very permeable and were visually detached from the original masonry due to incompatibility with the setting.

**Author Contributions:** For structure, M.L.L.C.; methodology, M.L.L.C., D.S.-A. and S.L.-A.; Analytica, M.L.L.C., D.S.-A. and S.L.-A.; validation, D.S.-A. and S.L.-A.; historical analysis, M.L.L.C. and I.d.P.; research, M.L.L.C.; resources, M.L.L.C.; data curation, M.L.L.C., D.S.-A., S.L.-A. and I.d.P.; writing—preparation of the original draft, M.L.L.C., D.S.-A., S.L.-A. and I.d.P.; writing, revision, and editing, M.L.L.C., D.S.-A., S.L.-A. and I.d.P.; translation: M.L.L.C.; visualization, M.L.L.C.; supervision, D.S.-A. and S.L.-A.; project administration, M.L.L.C.; funding acquisition, M.L.L.C. All authors have read and agreed to the published version of the manuscript.

**Funding:** The dissemination of the results of this research stage is financed by the International University of Ecuador the project UIDE-DGIP-MAT-PROY-22-005. This paper is part of the research of the doctoral program in Construction and Architectural Technology DTCA of the Polytechnic University of Madrid, whose subject is: *"Characterization of the mortar coating of Colonial Quito in the 16th, 17th and 18th centuries"*.

**Data Availability Statement:** The sources of historical books in this document are primary sources that have not been updated since the middle of the 20th century; some are considered *'gray literature'*, but they serve as a fundamental basis within the qualitative investigation of the present manuscript.

**Acknowledgments:** Thanks to the openness and ease of access and sampling of the church and its authority of the Church of Santo Domingo, especially to the Prior Gonzalo Suárez Carvajal, O.P. The analysis was carried out by the Geological Techniques Unit of the Support Center for Research in Earth Sciences and Archaeometry of the Complutense University of Madrid (Spain).

**Conflicts of Interest:** The authors declare no conflict of interest. Sponsors had no role in the design of the study; in the collection, analysis or interpretation of data; in the writing of the manuscript or in the decision to publish the results.

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
