# Peer review of "Characterization and Analysis of the Mortars of the Church of Santo Domingo in Quito (Ecuador)"

_heritage, doi:10.3390/heritage5040207_

Round 1

Reviewer 1 Report

Line 178: you can try to decrease the damage when extracting a mortar sample but is impossible to avoid any kind of damage on the building ! Therefore it is not correct to speak about “Non Destructive Testing”

Line 201: the term “thin section” instead of “thin film” should be used

Line 224: Figure 5 is not necessary  for the purposes of the research

Line 234: concerning Table 2, samples ISD 06, 07, 011, 12, 13, it is not comprehensive to say “Mortar based on volcanic aggregate”! The kind of binder should be added.

To this purpose,  in the Results section, an accurate petrographic description of the mortars belonging to each construction phases must be added. This description should examine the composition of the aggregate (type of single crystals, rock fragments), the grain morphology (sphericity, rounding), the grain size distribution, the type of binder, the possible presence of lime lumps, the binder/aggregate ratio. If the binder is not present, this must be highlighted.

Only through such study it is possible to correctly highlight the differences between the mortars of the various construction phases

Author Response

Dear reviewer

The reviewer is congratulated for the recommendations suggested to improve the comprehension and reading of the research.

The very large paragraphs have been restructured, both those described by the reviewer and the others that had the same problem, to make the research easier to read.

Line 178: The structure of the line and the paragraph regarding NDT was revised.

Line 201: Replaced with the correct term.

Line 224: Deleted Figure 5. And the numbering was changed for all the following figures.

Line 234: Table 2 was rectified.

Dear reviewer, we fully agree with your comment, but this research will be part of a complementary future work when designing the intervention materials.

Reviewer 2 Report

The authors performed a detailed mineral characterization using the mortar samples taken from the historic church of Santo Domingo. The results provide valuable information regarding the material composition and understanding of the date of the construction period. I suggest accepting the article. 

My only comments will be about Figure 3 - Its resolution is low and a bit unorganized. The authors may want to update the visual accordingly. 

Author Response

Dear reviewer

The reviewer is congratulated for the recommendations suggested to improve the comprehension and reading of the research.

Thank you very much for the positive comment on the research, Latin America and specially Ecuador does not have examples of research on mortars, and characterization of materials in their buildings with heritage value

Reviewer 3 Report

I think the manuscript needs a severe English revision.

I find sentences consisting of four or more lines all along the MS, e.g. P3 the paragraph L 103 -111; P9 L 268-274 etc.

I found the manuscript very complicated to understand.

I list below some of the reasons why I suggest that MS should be rejected.

Some terms are not appropriate for this topic, for example:

when the authors say dosage, do they mean binder/aggregate ratios (p3, L112)?

binocular magnifying glass, do they mean a Binocular Stereoscopic Microscope (P 6, L189)?

Thin film, do they refer to thin section (P6, L201)?

Polished section, do they refer to thin section (P7, L228)?

What do they refer with optical microscopy (P7, L228)?

Regarding the mortar preparation procedure is concerned, the method does not follow the standard process of separating the binder from the aggregates. Thus, the results can be questionable.

There are also other mistakes, such as that authors say by XRD the chemical characterization (P 6, L192) or textural relationship (P 7, L 237) can be established.

Conclusions

P10, L322-329. I don’t think that this conclusion can be extracted from the results.

P11, L348-349. If larnite is a phase present in all samples, why is it concluded that it is a synthetic material hypothetically added in the last decades?

Other weaknesses of the Ms:

I think that figure 3 should be changed.

The sampling description should be removed from the figure and included in the text, as the font size is too small for easy reading. Also, there should be homogeneity in the font size in the sample’s names and in sample ISD-06 the term phase does not appear in full.

Figure 4, once again, the font size in the figure is too small for easy reading.

Figure 5 has no place in a scientific article.

Figure 6 and 7, I am not sure if these figures correspond to secondary electron images, I think they correspond to images of backscattered electrons.

I also doubt that the magnifications of the images are correct (P8, L244 and 246; L250 and L252).

Regarding the references, I miss references from authors with wide experience in the study of mortars.

Author Response

Dear reviewer

The reviewer is congratulated for the recommendations suggested to improve the comprehension and reading of the research, and the structure and terminology in English have been revised.

The very large paragraphs have been restructured, both those described by the reviewer and the others that had the same problem, to make the research easier to read.

Line 103-111: Revised the paragraph structure.

Line 112: Replaced with the correct term.

Line 189: Replaced by the correct term.

Line 268-274: Revised the structure of the paragraph.

Line 178: The structure of the line and the paragraph regarding NDT was revised.

Line 201: Replaced with correct terminology.

Line 224: Removed Figure 5. And the numbering was changed for all the following figures.

Line 228: Restructured paragraph and replaced with correct terminology.

Line 234: Rectified Table 2.

Line 332-329: Deleted this paragraph.

Line 348-349: Due to a transcription error, the presence of "larnite" is indicated in all samples, but as can be seen in Table 1 it is only identified in sample ISD-04. The entire paragraph has been corrected in response to the reviewer's question.

Figure 4: The graphical part of this image, as well as the others in the document, has been improved.

Figure 5: Figure 5 was deleted as it did not generate a research contribution. And the numbering was changed for all the following figures.

Figure 6, 7, 8, 9: In the lower legend, the correct term was replaced by the correct value of the magnifications of the samples.

The bibliographic references of the document were revised.

Round 2

Reviewer 1 Report

Thank you for accepting the suggestions

Author Response

Dear reviewer

The reviewer is congratulated for the recommendations suggested to improve the comprehension and reading of the research.

Thank you very much for the positive comment on the research; Latin America and especially Ecuador do not have examples of research on mortars and characterization of materials in their buildings with heritage value.

Reviewer 3 Report

I find that there are still some major questions to be answered.

In general, the following issues need to be developed:

First: Considering a mortar is mixture of binder and aggregates, what are the nature of theses mortars?

Second: Was the use of volcanic aggregates to give mortars pozzolanic properties?

Was the use of volcanic aggregates intentionally?

Third:

The authors indicate that the presence of larnite is due to the use of Portland cement during restoration. Although, the use of Portland cement has not been very successful in the heritage conservation.

However, in the MS there are not a good characterisation of the original mortars in order to development of mortars with similar characteristics. I insist, a classic characterisation of mortars: binder/aggregate ratio, nature of the binder, grain size of the aggregates, intentionality in the use of a particular type of aggregates etc.

More specifically:

P 5, L173-176: Why has an EVS been carried out if after this data has not been used in this work?

For mineralogical characterization (P6, L192-192) a portion at 53mm was used.

Since there is no grain size analysis, what does the mineralogy obtained in that fraction correspond to? (Table1)

P7, L228-234: No results are reported, only analytic methods are indicated.

Table 2. when it says Lime joint mortar with volcanic aggregates, what does lime mortar mean?

The data given in the figure captions Figures 6, 7 and 8 should be described in the main text.

Once again, there are not results of the petrographic microscopy study.

P9, L284-285: I would like to assume that this statement has been made based on observations at higher magnifications than those in figures 6, 7, and 8.

Author Response

Dear reviewer

The paper has been subjected to your recommendations to improve the comprehension, readability, and impact of the research, with the parameters of the Heritage journal.

First

Generally, they are lime mortar as a binder and volcanic aggregates of the type pyroclastic trachyte of the Cotopaxi volcano. For reasons of conservation of the cultural property studied, it was not possible to sample in all the zones the aggregate + binder set. However, it has seemed to us positive and interesting to the scientific community that the inclusion of some samples could be sampled from which only relatively large fragments of volcanic aggregate could be sampled. The use of this aggregate has served to establish internal correlations between the different stages of construction, by composition and texture.

Second:

The aspect of the pozzolanic properties of the mortars has not been the purpose of this work; however, we find this comment very interesting and will address it in future studies.

Probably, the use of the volcanic material is due to economic issues and determination of the site because the religious order was designated a specific place to extract the building material in the seventeenth century given the proximity of the volcanic deposits, although some suitability for some finishes is documented. This hypothesis should be tested in future work.

Third:

The reviewer has right in all his comments, but the journal chosen 'Heritage' for this publication has a broad scientific scope. The study suggested by the reviewer will be the subject of another paper in a more specific journal of Materials or Civil Engineering applied to Architectural Heritage.

P 5, L173-176: Why has an EVS been carried out if after this data has not been used in this work?

Visual systematic evaluation (EVS) has been used as a basis for the analysis of sampling versus historical correlation of the place. It has been carried out with the corresponding data collection sheets. We did not consider it relevant to include the data sheets in the body of the article. 

For mineralogical characterization (P6, L192-192) a portion at 53mm was used. Since there is no grain size analysis, what does the mineralogy obtained in that fraction correspond to? (Table1)

The polycrystalline powder method requires a sample fraction with a grain size of less than 53 microns obtained from any other initial size by grinding and sieving.

P7, L228-234: No results are reported, only analytic methods are indicated.

The results are summarized in the figures, and the text referred to by the reviewer is a methodological clarification of the content of these figures.

Table 2. when it says Lime joint mortar with volcanic aggregates, what does lime mortar mean?

The identification of calcite in most of the samples has led to the conclusion that the initial binder was lime-based (probably hydrated lime, according to the local construction tradition).

The data given in the figure captions Figures 6, 7 and 8 should be described in the main text.

This recommendation will be followed in the final manuscript, taking care not to repeat it.

Once again, there are not results of the petrographic microscopy study.

P9, L284-285: I would like to assume that this statement has been made based on observations at higher magnifications than those in figures 6, 7, and 8.

The reviewer is correct, and these observations were made as indicated. A detailed petrographic description is beyond the broad scope for which the journal is intended.